# MindDETR: Beyond Semantics, Exploring Positional Cues from Brain Activity

## Abstract

Decoding visual stimuli from brain recordings offers a unique opportunity to understand how the brain represents the world and seeks to interpret the connection between computer vision models and our visual system. Recent efforts mainly adopt diffusion models to reconstruct images from brain signals. However, while these methods generally capture correct semantic information, they often struggle with precise object localization. Additionally, the commonly used proxy task, image reconstruction from brain signals, mainly measures semantic consistency, to some extent neglecting positional information of the decoded signals. In this work, to encourage more accurate brain signal decoding, we propose to use object detection as the proxy task, aiming at decoding both the semantic and positional cues from brain recordings. Based on this task, we propose MindDETR, a brain recording-based object detection model with the DETR pipeline. After aligning feature representations with a pretrained image-based DETR model, our model demonstrates that accurately brain decoding at both semantic and positional levels is feasible, and our detection-based approach achieves significantly superior results than existing reconstruction-based approaches. This result suggests the effectiveness of applying object detection as a proxy task for brain signal decoding. Our code will be publicly available.

## 1 Introduction

Decoding visual signals from brain activity is a fundamental challenge in neuroscience (Horikawa et al., 2013; Kay et al., 2008; Nishimoto et al., 2011). This research area not only allows us to explore the complex patterns of neural activity but also provides a bridge to link computer vision models and the human visual system. Recently, significant efforts (Takagi & Nishimoto, 2023; Scotti et al., 2024a; Ozcelik & VanRullen, 2023; Chen et al., 2023b; Luo et al., 2024; Wang et al., 2024a) have been made in this field, particularly in reconstructing visual stimulus images from functional magnetic resonance imaging (fMRI) signals. Deep models, especially diffusion-based models, have achieved promising progress in this task, producing reconstructed images with high semantic fidelity (see Figure 1-a).

However, despite considerable achievements, a major issue persists in this research line: the positional features of the decoded images are often inaccurate, even though their semantics are correct (see Figure 1-b). Besides, another noteworthy issue is that the leading methods in this task are all based on the diffusion model (Ho et al., 2020a; Rombach et al., 2022b), and the powerful diffusion model generally can generate realistic images with consistent semantics from a few cues, such as some keywords or a sentence (see Figure 1-c). This might provide a shortcut for the image reconstruction task, preventing the deep decoding of complex patterns in brain activity. Besides, the random process in diffusion models also results in the instability of decoding results (see Appendix A.2), which is unfriendly to neuroscience analysis.

In this work, we propose a new proxy task, object detection from brain signals, for the field of brain signal decoding. Same as object detection (Everingham, 2008; Zou et al., 2023), this task aims to estimate the semantic category and location coordinates of each object, and the only difference is the input of our task is brain recordings, *i.e.* fMRI signal, rather than RGB images. Compared to image reconstruction, this task aims to capture more detailed cues and evaluate the fidelity of semantic and positional decoding.

Figure 1: **Brain signal decoding.** Existing works generally use image reconstruction as the proxy task for fMRI signal decoding and get reasonable results with consistent semantics (*a*). However, these works are hard to recover the details accurately, such as numbers, sizes, or locations of the objects (*b*). The diffusion model can produce images with limited semantic information (*c*), potentially causing overfitting in related methods and hindering detailed brain activity decoding. To encourage further exploration in detailed decoding, we propose to use object detection as the proxy task (*left panel*) and show that detection from fMRI signals is feasible and promising (*d*).

However, brain activity is extremely complex, and fMRI can only record brain signals in a relatively coarse manner. In this context, whether accurate detection results can be obtained from such signals is a question worth exploring. To answer this question, we build our model based on the DETR pipeline (Carion et al., 2020), which is an end-to-end transformer-based object detection model. However, the fMRI signal is a noisy and coarse-grained recording of brain activity, and decoding detailed information from this type of data poses significant challenges. We find that the learned representations in pretrained image detection models offer strong priors for brain decoding, and applying feature distillation between the representations of fMRI and image can greatly improve decoding accuracy. In this way, we demonstrate that decoding both semantic and positional information from fMRI signals is feasible (see Figure 1-d) and is a promising research line for brain decoding.

Furthermore, to quantitatively compare the decoding fidelity between our model and existing image reconstruction-based models, we apply the same pretrained detection model (Liu et al., 2022) on the reconstructed images to get the detection results of other methods, and then compare these two kinds of works based on object detection metrics. Experimental results show that our method numerically outperforms reconstruction-based methods, and we also provide corresponding qualitative results for reference. These findings indicate that, compared to image reconstruction, using object detection as a proxy task for brain signal decoding has unparalleled advantages in decoding details of the visual stimuli. We hope that this proxy task will encourage further exploration of brain signal decoding within the research community.

To summarize, the contributions of this work are as follows:

- We observe that existing fMRI-to-image decoding models struggle to accurately recover positional information, which is partly because the image reconstruction task (and metrics) mainly focus on semantic consistency rather than the detailed information.

- We propose employing object detection as a proxy task for brain signal decoding and present that accurately decoding the details, such as the semantics, numbers, sizes, and locations of objects, from fMRI signals is feasible.

- We introduce a fMRI-based detection model which significantly outperforms reconstruction-based decoding baselines on the detection metrics, suggesting the effectiveness and superiority of the detection-based brain signal decoding schemes.

## 2 RELATED WORK

**Image reconstruction from brain signals.** In previous studies, considerable efforts have been made to reconstruct images from brain signals. The core of image reconstruction work involves extracting fMRI features aligned with visual image features from the fMRI signals, and using them to guide the generation model for image reconstruction. Early work used VCG networks to extract image features (Horikawa & Kamitani, 2017; Shen et al., 2019b), decode fMRI features aligned with image features

from fMRI, and subsequent work utilized GANs to reconstruct images from fMRI features(Shen et al., 2019a). The following work utilized a BigGAN-based image feature extractor to extract features, obtained fMRI features through ridge regression, and then used the fMRI features to fine-tune the image generation process of BigGAN (Mozafari et al., 2020; Ozcelik et al., 2022). Recently, diffusion models (Rombach et al., 2022b; Ho et al., 2020b; Rombach et al., 2022a) have become increasingly popular in the field of image generation, leading to methods that apply diffusion models to reconstruct images (Takagi & Nishimoto, 2023; Scotti et al., 2024a; Ozcelik & VanRullen, 2023; Chen et al., 2023b).

In addition to reconstructing high-resolution images, some studies also focus on reconstructing the corresponding textual information of the visual stimulus (Ferrante et al., 2023; Chen et al., 2023a; Han et al., 2023). The MinD-Video (Chen et al., 2024) attempts to reconstruct visual stimuli in video format from contiguous fMRI signal frames. Recently many studies have explored aligning fMRI signals from different subjects in order to train a unified model (Wang et al., 2024a; Gong et al., 2024; Bao et al., 2024; Wang et al., 2024b; Xia et al., 2024; Liu et al., 2024; Zhou et al., 2024; Thual et al., 2023; Scotti et al., 2024b).

Different from previous works, we propose using object detection as a proxy task, with our focus on decoding both semantic and spatial information from fMRI signals simultaneously.

**Object detection.** As a cornerstone in computer vision, object detection aims to identify the objects within the given image by predicting their semantic labels and bounding boxes. *This task captures the numbers, semantic labels, sizes, and locations of the objects, which is an ideal proxy task for brain decoding.* Specifically, the methods in this field can be broadly divided into three groups, one-stage methods (Redmon et al., 2016; Liu et al., 2016; Duan et al., 2019), two-stage methods (Girshick et al., 2014; Girshick, 2015; Ren et al., 2015), and attention-based methods (Carion et al., 2020; Zhu et al., 2020; Liu et al., 2022). Among these works, the attention-based models avoid the need for complex object detection designs, *e.g.* anchors and proposals, simplifying the encoding and detection processes through attention mechanisms. Furthermore, the attention patterns learned by such methods provide valuable insights in explaining brain activities. Therefore, we build our model based on the attention-based pipeline.

**Knowledge distillation.** Knowledge distillation (KD) (Hinton et al., 2015) is originally designed for transferring the encapsulated knowledge, represented by predicted logits, from a well-trained teacher model into a student model with less representational capacity. This idea is extended to the feature level by several works (Heo et al., 2019; Park et al., 2019; Zong et al., 2022), and the feature distillation is generally used to align the feature representation between different modalities (Chong et al., 2021; Chen et al., 2022; Wang et al., 2023). In this work, we find that directly applying detection algorithms on fMRI signals is hard to decode detection results accurately, while aligning the features between image data and fMRI data is a good way to improve the decoding accuracy of brain activity.

## 3 APPROACH

### 3.1 OVERVIEW

As shown in Figure 2, we build our model based on the DETR pipeline. Specifically, our model first uses a linear adapter to align the fMRI features from different subjects into a shared space (Section 3.2). Then we use a light-weight backbone, consisting of a Multi-Layer Perceptron (MLP) and a convolutional (CONV) layer, to extract the features (Section 6), followed by a standard encoder-decoder structure to get the detection results from the queries (Section 3.4). Besides, we also leverage a pretrained model to provide additional supervision in two feature levels (Section 3.5), which is removed in the inference phase.

### 3.2 LINEAR ADAPTER

As reported in Wang et al. (2024a); Scotti et al. (2024b), due to the significant variability in fMRI signals across different subjects, cross-subject neural signal decoding is a challenging problem. However, training a separate model for each subject makes it difficult to discover general patterns in neural activity and also faces the issue of insufficient training data. To address this issue, we design a

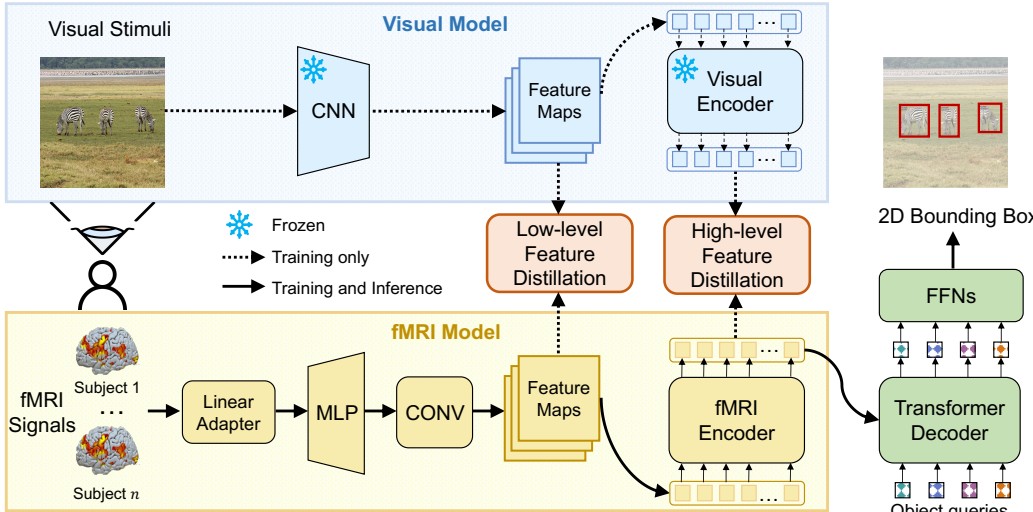

Figure 2: **Overview of MindDETR.** We build our model based on the DETR pipeline. The teacher model (*top lane*) takes images as input and is only involved in the training phase, where its parameters are frozen. The student model (*bottom lane*) takes fMRI signals as input, with additional feature supervision from the fixed teacher model, and generates object detection bounding boxes for corresponding visual stimuli images in the inference phase.

linear adapter to align fMRI signals from different subjects into a common space, thereby enabling training on mixed data from multiple subjects. Specifically, suppose there are $n$ subjects, and the fMRI signal from the $i$-th subject is represented as $s_i \in \mathbb{R}^{\text{len}_i}$, $i = 1, 2, \ldots, n$, where $\text{len}_i$ denotes the dimension of the recording features for subject $i$. Note the feature dimension is generally different for different subjects, and we orderly concatenate the features of each subject and then mask those that do not belong to $s_i$ to get the features of subject $s$ with a fixed dimension. This process can also be regarded as a kind of position-aware zero-padding. After that, we use a shared linear layer to map the features of different subjects into a unified latent space.

## 3.3 FEATURE EXTRACTION

The backbone module extracts features from the fMRI signals in the shared latent space. Considering the relatively low resolution in fMRI signals, any method focused on extracting local features may result in information loss. Therefore, we directly employ a MLP for brain signal feature extraction, with the same structure used in MindEye (Scotti et al., 2024a). To enable the feature distillation (Section 3.5), we set the output dimension of the MLP to $H \cdot W$ and reshape the resulting 1-D features into 2-D features. Furthermore, we observe that the feature maps extracted by the MLP present limited representation capacity. Therefore we further add a two-dimensional CONV layer (kernel = K, padding = $(K + 1)/2$, stride = 1) after the MLP to further extract local features. We also find the kernel size contributes to the final accuracy significantly, and the related experiments are reported in Section 4.4.

## 3.4 ENCODER AND DECODER

For the encoder and decoder components, we retain the design from DETR (Carion et al., 2020). The initial DETR model has a slow convergence speed, and DAB-DETR (Liu et al., 2022) addresses this issue by concretizing the abstract learnable object queries into anchors to accelerate convergence. We follow this design in our model. Further, some DETR-related works enhance DETR's performance for small object detection, such as Deformable DETR (Zhu et al., 2020) which introduces deformable attention in the encoder. However, we find this optimization unsuitable for the brain detection task due to the inability to extract multi-scale feature maps from brain signals, which are crucial for such optimization methods.

## 3.5 FEATURE DISTILLATION

Due to the high noise levels in fMRI signals and the intricate patterns of neural activity, directly predicting detection results from fMRI signals is challenging. We find the feature representations learned by pre-trained image detectors could offer robust semantic priors for fMRI models, so we utilize these features to provide additional supervision for our model in the feature space. We refer to the visual model and our fRMI-based model as the teacher and student respectively in this part. Specifically, we leverage the pretrained DAB-DETR (Liu et al., 2022) with ResNet-50 backbone (He et al., 2016) as our teacher model, and the feature supervisions are applied in two feature levels, *i.e.* low-level feature distillation and high-level feature distillation.

The low-level distillation is conducted on the backbone features, where the feature shape of the teacher model is $C^T \times H \times W$. To ensure alignment of the feature shape, we modify the MLP in our backbone to produce a 1D feature with dimensions HW, which is then reshaped into a 2D feature before being fed into the 2D convolutional layer. Then we apply feature-based KD between the teacher feature $F^T$ and student feature $F^S$ with L2 norm:

$$\mathcal{L}_{\text{low}} = \frac{1}{C^T HW} \|F^T - \phi(F^S)\|_2^2, \tag{1}$$

where $\phi$ is a learnable projector to transform the student's feature into $C^T$ dimensions. Besides, the projector also benefits to model's representation capability (Chen et al., 2020). Additionally, the high-level distillation is performed on the token sequences outputted by the encoders. Similarly, we also apply the L2 norm between teacher tokens $F'^T$ and $F'^S$:

$$\mathcal{L}_{\text{high}} = \frac{1}{C'^T L} \|F'^T - \phi'(F'^S)\|_2^2, \tag{2}$$

where $C'^T$ and L denote the feature dimensions and length of the teacher tokens. $\phi'$ is the learnable projector for the token sequence.

## 3.6 TRAINING OBJECTIVE

Same as DETR, we use the set prediction loss to optimize our model and denote the object detection loss as $\mathcal{L}_{\text{det}}$. Therefore, our final loss can be formulated as:

$$\mathcal{L} = \mathcal{L}_{\text{det}} + \lambda_1 \mathcal{L}_{\text{low}} + \lambda_2 \mathcal{L}_{\text{high}}, \tag{3}$$

where $\lambda_1$ and $\lambda_1$ are the weight coefficients to balance these loss terms.

## 4 EXPERIMENTS

### 4.1 TASK SETUPS

**Data preprocessing.** We use the Natural Scene dataset (NSD) dataset (Allen et al., 2022) for our experiments. During NSD's data collection, the images used as visual stimuli are from a subset of the COCO dataset `train` and `val` sets (Lin et al., 2014). These images were cropped and resized to $425 \times 425$ pixels before being presented to the subjects (see Appendix A.1 for more dataset details). To facilitate fMRI-based object detection, we adjusted the COCO bounding box annotations to correspond with the resized images. Additionally, objects that lost over 90% of their area during resizing were removed due to information loss, which resulted in the removal of approximately 10.3% of the annotations. Objects occupying less than 0.1% of the resized image area were also excluded, eliminating an additional 10.1% of the annotations.

**Data splits.** For the NSD dataset split, we followed the method of Takagi. (Takagi & Nishimoto, 2023), using data from subject 1, 2, 5, and 7 only. The test set consisted of 982 images that were viewed by all four subjects along with the corresponding fMRI signals, while the remaining data were used for the training. For the fMRI signals, we used voxels from the region of "nsdgeneral", which is a general ROI manually drawn on "fsaverage" data, covering voxels responsive to the NSD (Allen et al., 2022) experiment in the posterior aspect of the cortex.

**Hyperparameters.** We set the dimension of the fMRI joint space to $H = 4096$, the convolutional kernel size to $K = 5$, and the number of feature channels to 64 (with the number of feature channels

Table 1: The results of the Average Precision (AP) and Average Recall (AR) metrics.

| Object Type | Method | Input | Average Precision ↑ | | | Average Recall ↑ | | |
|---|---|---|---|---|---|---|---|---|
| | | | $AP_{30}$ | $AP_{50}$ | $AP_{70}$ | $AR_{30}$ | $AR_{50}$ | $AR_{70}$ |
| Small | DAB-DETR (Liu et al., 2022) | Images | 71.20 | 60.70 | 39.10 | 95.40 | 86.60 | 58.10 |
| | Takagi. (Takagi & Nishimoto, 2023) | fMRI | 0.10 | 0.00 | 0.00 | 2.30 | 0.50 | 0.00 |
| | MindEye (Scotti et al., 2024a) | fMRI | 0.47 | 0.23 | 0.08 | 6.12 | 1.93 | 0.40 |
| | MindDETR (Ours) | fMRI | **2.70** | **0.40** | **0.10** | **17.50** | **5.98** | **1.15** |
| Medium | DAB-DETR (Liu et al., 2022) | Images | 85.60 | 81.10 | 70.00 | 98.50 | 96.20 | 86.40 |
| | Takagi. (Takagi & Nishimoto, 2023) | fMRI | 0.60 | 0.00 | 0.00 | 7.50 | 1.90 | 0.00 |
| | MindEye (Scotti et al., 2024a) | fMRI | 3.03 | 0.55 | 0.03 | 20.30 | 6.88 | 1.03 |
| | MindDETR (Ours) | fMRI | **11.50** | **4.10** | **1.00** | **47.38** | **24.05** | **5.10** |
| Large | DAB-DETR (Liu et al., 2022) | Images | 90.20 | 86.60 | 77.90 | 99.90 | 99.10 | 95.70 |
| | Takagi. (Takagi & Nishimoto, 2023) | fMRI | 4.20 | 2.30 | 0.60 | 29.50 | 18.10 | 8.80 |
| | MindEye (Scotti et al., 2024a) | fMRI | 17.40 | 9.65 | 3.53 | 56.35 | 38.52 | 19.65 |
| | MindDETR (Ours) | fMRI | **26.20** | **18.90** | **10.10** | **76.43** | **62.45** | **34.10** |
| All | DAB-DETR (Liu et al., 2022) | Images | 81.90 | 76.00 | 62.40 | 97.90 | 93.40 | 80.30 |
| | Takagi. (Takagi & Nishimoto, 2023) | fMRI | 1.60 | 0.80 | 0.30 | 14.70 | 8.00 | 3.70 |
| | MindEye (Scotti et al., 2024a) | fMRI | 7.47 | 4.12 | 1.57 | 28.43 | 17.25 | 8.20 |
| | MindDETR (Ours) | fMRI | **12.90** | **8.50** | **4.50** | **48.40** | **31.85** | **14.47** |

for the teacher model set to 256). The hyperparameters in Equation 3 are set as $\lambda_1 = 2$ and $\lambda_2 = 60$ (see Appendix A.4 for the ablation). For training, we utilize the AdamW (Loshchilov & Hutter, 2017) optimizer, and both the learning rate and weight decay are set to $10^{-4}$. The model is trained on 4 NVIDIA A100 GPUs for 50 epochs with a batch size of 8 per GPU.

**Evaluation metrics.** We apply modified evaluation metrics of COCO for matching the brain object detection task. Specifically, for Large, Medium, Small, and All objects in the COCO evaluation metrics, under different IoU (Intersection over Union) thresholds, we compute the AP (Average Precision) and AR (Average Recall) for each category and take the average as the result. Here, we set the IoU thresholds to 30, 50, and 70, relaxing the strict localization requirements and resulting the metrics of $AP_{30}$, $AP_{50}$, $AP_{70}$, $AR_{30}$, $AR_{50}$, $AR_{70}$ accordingly.

**Baselines.** Given the lack of current research on brain target detection, we selected two representative image reconstruction works, Takagi (Takagi & Nishimoto, 2023) and MindEye (Scotti et al., 2024a), to demonstrate the effectiveness of our method. To ensure comparability, we standardized the dimensions of the reconstructed images from these methods and conducted object detection on these reconstructed images. The detected results can be regarded as the *theoretical upper bound* of these reconstruction-based baseline models. For object detection, we used the pretrained DAB-DETR model (Liu et al., 2022) provided by MMDetection (Chen et al., 2019). As a reference, we also present the detection results of DAB-DETR using standard visual images as inputs.

## 4.2 MAIN RESULTS

To assess performance, we conduct extensive experiments and summarize the results in Table 1. Based on these results, we can get the following conclusions:

- First, the proposed method is significantly superior to the reconstruction-based methods across various settings, which confirms the effectiveness of the proposed method. For example, MindDETR surpasses MindEye by 8.47 on $AR_{30}$ and 27.08 on $AR_{30}$, respectively.

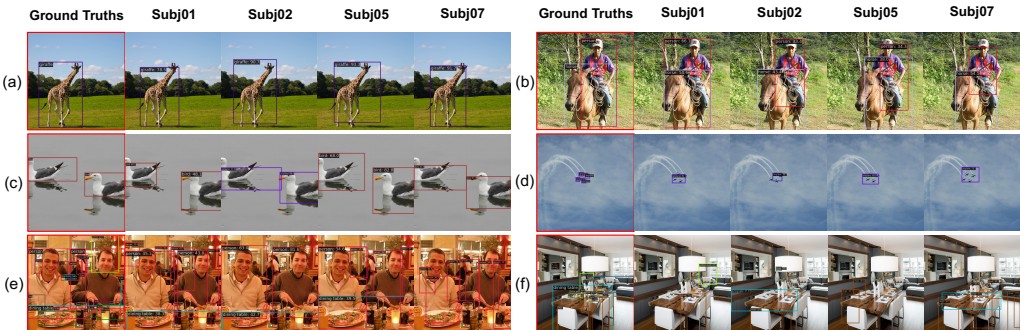

Figure 3: **Comparison of MindDETR and other image reconstruction models.** MindDETR only outputs the 2D bounding box in the results, while other models only output images, whose bounding boxes are generated by the 2D object detection method DAB-DETR (Liu et al., 2022).

Figure 4: **Comparison of results for different subjects with the same visual stimulus.** MindDE-TRcan get consistent detection results among different objects.

- Second, MindDETR achieves promising results (especially for the AR metric), which demonstrates that accurately decoding the locations of objects from the noisy and complex brain recordings is possible.
- Third, we find the reconstruction-based methods are hard to get satisfactory performance on brain detection, which suggests previous methods neglect the importance of spatial cues decoding and demonstrate the necessity of the proposed task.
- Finally, we also find there is still a lot of room for improvement in fMRI-based detection models, and we hope this study can facilitate this research line.

Overall, these findings illustrate the superior efficacy of MindDETR in enhancing fMRI-based object recognition performance and suggest object detection is a reasonable proxy task for brain signal decoding.

### 4.3 VISUALIZATION AND ANALYSIS

**Comparison of the qualitative results.** To further demonstrate the superior performance of the proposed method, we provide the the visualizations of proposed framework against other reconstruction-based methods in Figure 3. In particular, one can see that while both existing reconstruction-based methods and our method can preserve semantics well, reconstruction-based methods struggle to accurately restore the number and specific positions of objects. This indicates that these methods may primarily focus on semantic information while neglecting other important details, leading to poor performance. In contrast, our method has a better capability for decoding details. Additionally, the proposed model can detect visually distinct small objects (consistent with the results shown in Table

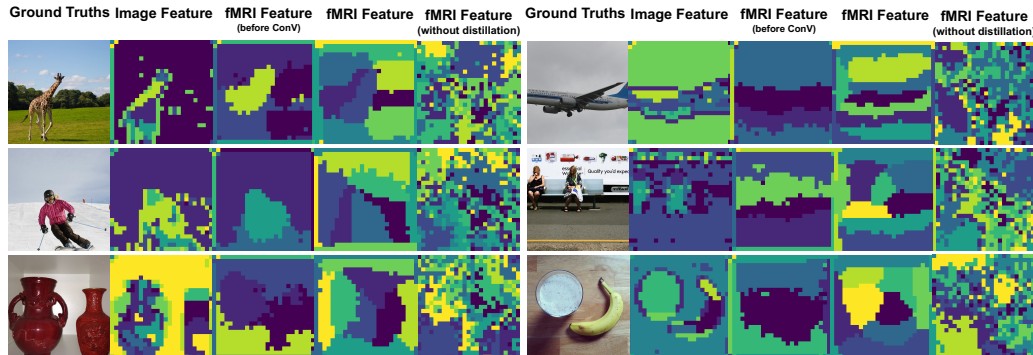

Figure 6: **Visualization results after clustering the feature maps.** The number of cluster centers is set to 10 with the K-means algorithm, and features belonging to the same cluster are colored identically. The image feature, fMRI feature (before CONV), fMRI feature, and fMRI feature (without distillation) respectively represent the clustering results of the feature maps from the teacher model's CNN, MindDETR's MLP, MindDETR's CONV, and the output of MindDETR's CONV trained without distillation.

1), which further demonstrates that our method surpasses reconstruction-based methods in detailed decoding.

**Consisteny among different objects.** Some studies (Wang et al., 2024a; Han et al., 2024) reported that existing brain decoding methods are hard to obtain consistent results between different objects, due to the complexity and distinction of neuro-activity. In Figure 4, we show that our method can maintain consistency in semantics, location, and quantity in most cases for the brain detection results of different subjects with the same visual stimulus images, except when there are extremely small multiple targets. This demonstrates the effectiveness and potential of our method. Besides, the mainstream reconstruction-based methods are based on the diffusion model with a random process, which may get instability results with the same object (refer to Appendix A.2 for more details).

**Pattern analysis.** To further investigate the differences between the proposed model, MindEye, and DAB-DETR, we show the category-wise results of these three models in Figure 5. In particular, we can find that *(i)*: the proposed MindDETR performs better than MindEye, and the performance of these two models exhibits a clear linear correlation. *(ii)*: DETR gets significantly better results than MindDETR and MindEye, which suggests there is still lots of improvement room for brain decoding models. *(iii)*: DETR shows distinctly different patterns with brain-based methods, *e.g.* 'parking meter' objects are easily captured by image detectors, but gener-

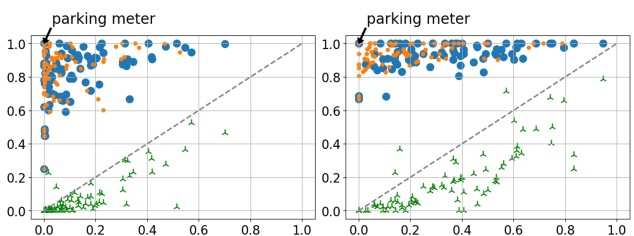

Figure 5: **Performance scatter plot.** The coordinates $(x, y)$ of a point denotes the performance of (Model A, Model B) of the same category, measured by $AP_{50}$ (*left*) and $AR_{50}$ (*right*). Blue, orange, and green points represent (MindDETR, DAB-DETR), (MindEye, DAB-DETR), and (MindDETR, Mind-Eye), respectively.

ally ignored by brain-based methods. This difference can be attributed to the attentional activities of the subjects during the data collection process, and analyzing such results contributes to a better exploration of neuroscience. See Appendix A.3 for the qualitative results and more analysis.

## 4.4 ABLATION STUDY

**Ablations on feature distillation and kernel size.** Table 2 presents the ablation study results with AP metric, examining the impact of varying kernel sizes (abbreviated as K) and the presence of low-level and high-level distillation modules. The baseline with a $5 \times 5$ kernel and no distillation

Table 2: **Ablation results** on feature distillation and kernel size. Experiments are conducted based on the Average Precision (AP) metrics. "K" represents the kernel size ("CONV" in Figure 2).

| | Distillation | | Large | | | Medium | | | Small | | | All | | |
|---|---|---|---|---|---|---|---|---|---|---|---|---|---|---|
| K | Low-level | High-level | $AP_{30}$ | $AP_{50}$ | $AP_{70}$ | $AP_{70}$ | $AP_{50}$ | $AP_{70}$ | $AP_{30}$ | $AP_{50}$ | $AP_{70}$ | $AP_{30}$ | $AP_{50}$ | $AP_{70}$ |
| 5×5 | ✗ | ✗ | 20.20 | 12.90 | 6.10 | 7.50 | 3.00 | 0.10 | **4.40** | 0.60 | 0.00 | 9.40 | 5.60 | 2.70 |
| 5×5 | ✓ | ✗ | 25.70 | 17.40 | 7.00 | 8.60 | 1.90 | 0.30 | 3.50 | 0.80 | 0.10 | 11.90 | 7.10 | 3.10 |
| 5×5 | ✗ | ✓ | 25.60 | 17.60 | 7.80 | 8.40 | 2.40 | 0.70 | 3.30 | 1.00 | 0.00 | 12.10 | 7.30 | 3.30 |
| 1×1 | ✓ | ✓ | 25.00 | 17.30 | 7.00 | 7.30 | 2.40 | 0.40 | 2.70 | 0.40 | 0.00 | 11.30 | 7.40 | 3.20 |
| 3×3 | ✓ | ✓ | 26.00 | 18.80 | 7.60 | 10.20 | **4.40** | **1.80** | 3.90 | 0.80 | 0.10 | 12.40 | 8.00 | 3.70 |
| 5×5 | ✓ | ✓ | 26.20 | 18.90 | **10.10** | **11.50** | 4.10 | 1.00 | 2.70 | 0.40 | **0.10** | 12.90 | 8.50 | **4.50** |
| 9×9 | ✓ | ✓ | **28.60** | **21.40** | 8.90 | 9.30 | 3.60 | 0.60 | 4.30 | **1.60** | 0.10 | **13.50** | **8.70** | 3.80 |

exhibits the lowest performance, especially for Small objects. Introducing low-level distillation with the same kernel size improves performance, particularly for Large objects ($AP_{30}$ = 25.70). High-level distillation alone further enhances results, slightly outperforming the low-level configuration. Using a 1×1 kernel with both distillation modules results in moderate gains, indicating smaller kernels benefit less from distillation. The $3 \times 3$ kernel with both distillations provides balanced improvements, notably for Medium objects. The $5 \times 5$ kernel with both distillations achieves the highest $AP_{70}$ for Large objects (10.10) and demonstrates strong overall performance. The $9 \times 9$ kernel with both distillations yields the best results overall, particularly for Large objects ($AP_{30}$ = 28.60, $AP_{50}$ = 21.40) and the aggregate ($AP_{30}$ = 13.50). Overall, Table 2 highlights the role of kernel size and demonstrates the effectiveness of low-level and high-level distillation in enhancing detection performance across different object sizes.

**Feature visualization of the ablation.** As explained in Section 3, distillation imposes spatial constraints on fMRI feature maps by aligning them with visual features. To visualize this and the enhancement effect of convolutional layers on feature maps, we further applied K-means clustering to the feature maps, as shown in Figure 6. Based on these visualizations, we can find: (*i*): fMRI feature (without distillation) is highly noisy, which suggests accurately decoding the details from fMRI signals is very challenging. (*ii*): the distillation-enhanced feature map contains positional information corresponding to the image features, which confirms the visual features can be used as effective guidance for brain decoding. (*iii*): the "CONV" module also brings inductive bias and enhances the discriminative of the features (especially for the objects and backgrounds).

## 5 CONCLUSION

**Summary.** We discovered that in previous brain signal decoding efforts with image reconstruction, the semantic content of images could be well reconstructed, but these methods did not perform well in accurately restoring objects' locations. To encourage further exploration in brain decoding, we proposed a proxy task, brain detection, aiming at simultaneously decoding semantic and positional information from brain signals. For this task, we designed MindDETR, a brain detector based on the DETR architecture, and used knowledge distillation to transfer knowledge from a visual object detector to the brain object detector. Our method significantly outperformed image reconstruction-based methods in terms of AP (Average Precision) and AR (Average Recall) metrics, further demonstrating that brain detection is a feasible and promising research direction.

**Limitations and Future Work.** First, while this work demonstrates the feasibility of performing object detection from brain signals, the performance of our model still lags significantly behind its theoretical limits. Furthermore, the process of reconstructing objects in images from brain signals lacks pixel-level interpretability. To address these two issues, we hope this task will spur further research, driving improvements in decoding accuracy with efforts from the whole community. Additionally, we plan to explore the interplay between image reconstruction and object detection tasks to enhance our understanding of brain signal decoding.

**Broader Impacts.** This work focuses on decoding the visual stimuli from human activity. While this task like mind reading may raise concerns about privacy breaches, actually the negative impact on society is negligible for several reasons. First, the current accuracy of such research is low. Second, it is challenging to apply related technologies to untrained subjects. Finally, access to private brain signals without permission is impossible.

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

# A  APPENDIX

## A.1  MORE DETAILS OF THE NSD DATASET

In the NSD dataset, we utilized the standard-resolution 1.8-mm fMRI data, stored as a three-dimensional matrix $A \in \mathbb{R}^{n \times m \times h}$. Only a portion of the voxels in this matrix contain actual values, while the rest are filled with zeros. "nsdgeneral" refers to a collection of visually relevant voxels in $A$ associated with the NSD experiment. In our study, we used only these marked voxels. Since these voxels are not contiguous within $A$, we extracted and rearranged them into a one-dimensional vector $B \in \mathbb{R}^w$. For subj01, subj02, subj05, and subj07, the lengths of $B$, *i.e.* $w$, are 15724, 14278, 13039, and 12682, respectively, with a total length of 55723.

In the NSD experiment, each subject views an image for 3 seconds, with approximately 9000 to 10000 different images presented. Each image is presented to a subject 2 to 3 times at different times. After data processing, we obtain 22000 to 30000 fMRI responses. Of these, 982 images were shown to subj01, subj02, subj05, and subj07 simultaneously. We used these images and their corresponding fMRI responses as the test set. During testing, we averaged the multiple fMRI responses for each image to obtain a unique fMRI response, ensuring that only one detection result is produced per image. The remaining images and their fMRI responses were used as the training set. During training, each image and its corresponding fMRI response were treated as independent data points.

## A.2  UNCERTAINTY IN IMAGE RECONSTRUCTION METHODS WITH DIFFUSION MODELS

Due to the complex patterns in brain activities and the noisy signal recordings, recovering the semantics of stimulus images is relatively easy to achieve, while accurately reconstructing the secondary features, such as locations, remains challenging (high uncertainty). Additionally, the design of diffusion models causes the generated images to differ in multiple runs with the same input. For these two reasons, diffusion-based methods generally produce varying object locations across multiple runs. To empirically support this, we used MindEye (Scotti et al., 2024a), a diffusion-based model for brain decoding, to generate multiple images with the same input. As shown in Figure 7, we can find the position (and size, quantity, etc.) of the object changes significantly in multiple image reconstructions. These results demonstrate that, to some extent, diffusion can lead to inaccuracy in the positioning of objects.

## A.3  DETAILS RESULTS FOR PATTERN ANALYSIS

Different from image reconstruction, the proposed task explicitly presents the decoding results with the bounding box and confidence. This kind of representation can more clearly reflect the attention tendency of the subjects during data recording, which helps to better analyze the activity patterns of the human brain. In Figure 4, we provide some qualitative results as examples. We can find the parking meters (pink bounding boxes) in the images have distinct visual features and are easily captured by image detectors, but they are often overlooked in brain signal detection. This is because subjects tend to focus more on foreground objects and ignore background information. By studying similar phenomena, we can easily identify the primary focus of subjects during data collection and further analyze attention patterns in human brain activity, which may contribute to future advancements in neuroscience. In contrast, it is difficult to achieve for image reconstruction models.

## A.4  ABLATION ON HYPERPARAMETERS $\lambda_1$ AND $\lambda_2$

We set $\lambda_1 = 2, \lambda_2 = 60$ in default in our implementation. The reason for this is to balance the losses of the three parts (*i.e.* $L_{det}$, $\lambda_1 L_{low}$ and $\lambda_2 L_{high}$ in Equation 3) so that they are similar in magnitude after multiplying by their respective weights, which is a common practice. Here, we provide additional experiments: we amplify or reduce the hyperparameters by a factor of 10 and show the ablation results in Table 3:

We can find the initial parameter choices of $\lambda_1 = 2, \lambda_2 = 60$ appear to be relatively optimal. An interesting phenomenon is that reducing the values of these two parameters leads to a degradation in performance, while increasing them slightly raises the $\text{AP}_{30}$ (low IoU constraint), but slightly

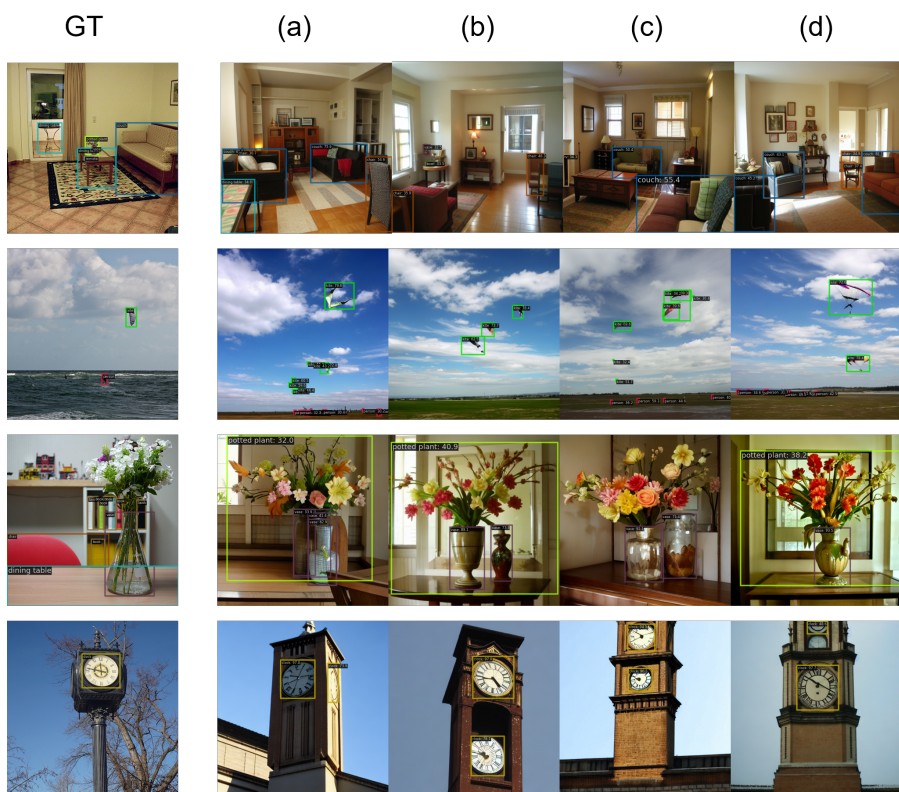

Figure 7: **Results of MindEye for multiple runs with the same input.** We can find the position (and size, quantity, *etc.*) of the object changes significantly in multiple image reconstructions.

Table 3: **Ablation on the hyperparameters** $\lambda_1, \lambda_2$ used in Equation 3. $*$ represents the default setting in our implementation.

| $\lambda_1$ | $\lambda_2$ | $AP_{30}$ | $AP_{50}$ | $AP_{70}$ |
|---|---|---|---|---|
| 2* | 60* | 12.9 | **8.5** | **4.5** |
| 2 | 6 | 12.0 | 7.0 | 2.7 |
| 0.2 | 60 | 12.9 | 7.9 | 3.3 |
| 0.2 | 6 | 11.7 | 6.2 | 2.0 |
| 2 | 600 | 13.0 | 7.9 | 4.0 |
| 20 | 60 | 12.4 | 7.7 | 3.5 |
| 20 | 600 | **13.5** | 8.0 | 3.9 |

decreases the $AP_{50}$ and $AP_{70}$ (under high IoU constraints). We believe that further fine-tuning of the hyperparameters could lead to a slight improvement in performance. However, the significance of this is not very great.

## A.5 ERROR CASES

To better illustrate the challenges in neural spatial decoding, we have summarized and visualized several prominent decoding errors in Figure 9.

• Case (a): Small objects like spoons, knives, donuts, and ovens were not detected, and persons appearing at the image edges were sometimes missed by some subjects. This raises the question of whether detection on the NSD dataset should include such small objects. Decoding fine details that subjects are less likely to notice from MRI signals is inherently challenging. By contrast, decoding central objects appears to be easier, while edge objects are more prone to being overlooked.

GT      DABDETR      MindDETR      MindEye

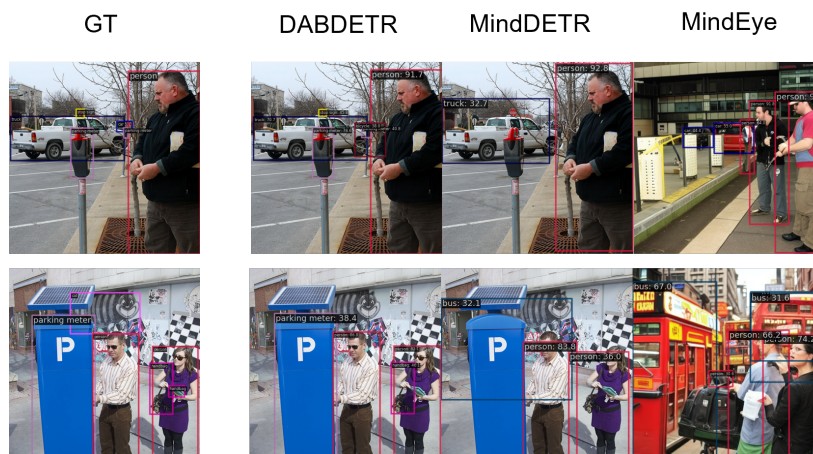

Figure 8: **Comparison of qualitative results.** We show the qualitative results of DAB-DETR (image-based detection model) and MindDETR (fMRI-based detection model). We also present the results of MindEye (fMRI-based reconstruction model) for reference. We can find the parking meters (pink bounding boxes) in the images have distinct visual features and are easily captured by image detectors, but they are often overlooked in brain signal detection. This is because subjects tend to focus more on foreground objects and ignore background information. By studying similar phenomena, we can analyze the attention patterns in human brain activity, which may contribute to future advancements in neuroscience.

• Case (b): A clock at the center of the image was not detected due to weak visual stimulation. The primary focus of the visual stimulus was architecture, not the clock. Since the clock category often functions as a non-core semantic object, it may be worth reconsidering whether this category should be included in the detection process.

• Case (c): A prominent object, such as a kite, was sometimes not detected, which is unexpected as it should be noticed by all subjects. This variability suggests that the model's learning for this category remains inadequate, potentially due to differences in how subjects perceive or prioritize the kite.

• Case (d): The detection of a person in the background was highly blurred, indicating that background objects are harder to detect, even though they may not pose a challenge for image-based detection. Similarly, the core semantic object, a baseball bat, was not detected, likely due to its small size.

In summary, objects located in the background, at the edges of images, or with small sizes tend to be more difficult to detect.

## A.6 RESULTS OF RECONSTRUCTION-BASED METHODS BASED ON DIFFERENT DETECTORS

For fairness, we have tested the performance of different object detection models on the reconstruction results, as shown in Tab. 4. We have incorporated two additional models, DINO (Zhang et al., 2022) and CO-DINO (Zong et al., 2023), for detecting the reconstructed images, which exhibit better detection performance on small objects.

Based on the experimental results, the choice of detection model has a noticeable impact on the results. However, a stronger detection model does not always yield better performance and may even cause slight metric regression. We hypothesize that this is because stronger detection models are more likely to identify blurred objects in the reconstructed images generated by MindEye, which weaker models might overlook. The inaccurate positional information associated with these blurred objects could contribute to the observed decline in metrics.

## A.7 RESULTS ON DIFFERENT ROIs.

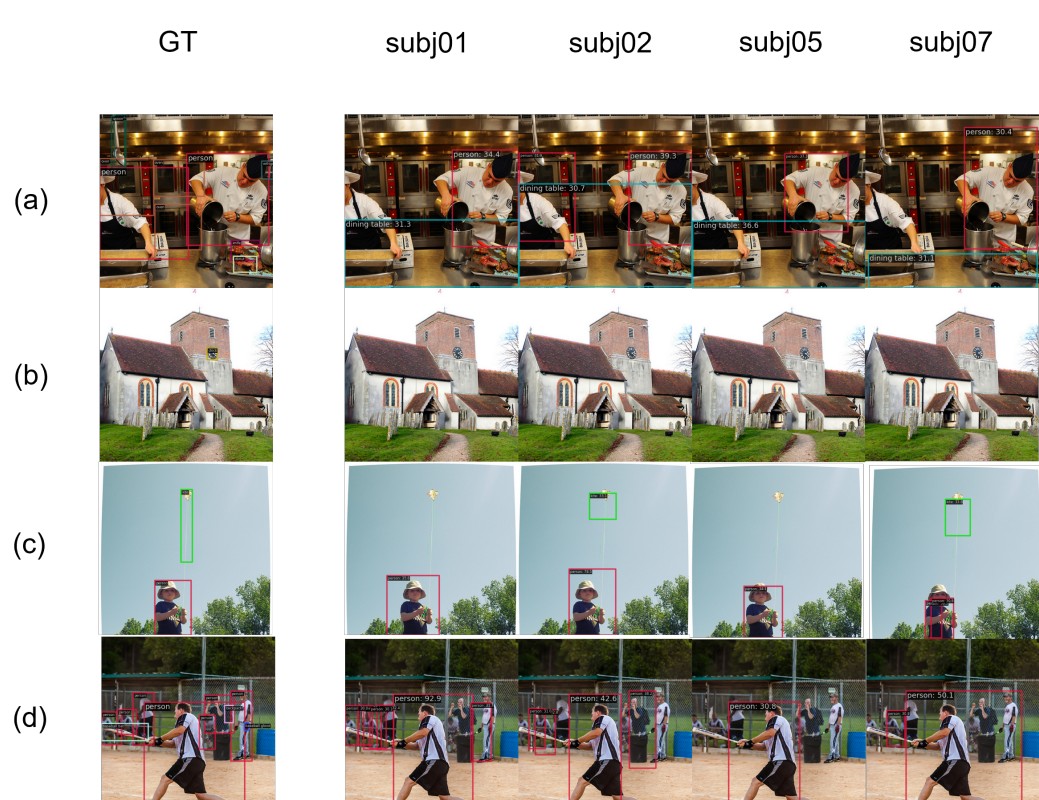

GT     subj01     subj02     subj05     subj07

(a)

(b)

(c)

(d)

Figure 9: **Error case visualization.**

Table 4: **The results of detection on reconstruction by different models.**

| Object Type | Method | Input | Average Precision ↑ | | | Average Recall ↑ | | |
|---|---|---|---|---|---|---|---|---|
| | | | $AP_{30}$ | $AP_{50}$ | $AP_{70}$ | $AR_{30}$ | $AR_{50}$ | $AR_{70}$ |
| Small | MindEye+DAB-DETR | fMRI | 0.47 | 0.23 | 0.08 | 6.12 | 1.93 | 0.40 |
| | MindEye+DINO | fMRI | 0.27 | 0.23 | 0.08 | 4.12 | 1.63 | 0.30 |
| | MindEye+CO-DINO | fMRI | 0.47 | 0.23 | 0.08 | 4.12 | 1.73 | 0.30 |
| Medium | MindEye+DAB-DETR | fMRI | 3.03 | 0.55 | 0.03 | 20.30 | 6.88 | 1.03 |
| | MindEye+DINO | fMRI | 2.33 | 0.45 | 0.03 | 17.40 | 5.88 | 0.83 |
| | MindEye+CO-DINO | fMRI | 2.53 | 0.45 | 0.03 | 19.20 | 6.58 | 0.73 |
| Large | MindEye+DAB-DETR | fMRI | 17.40 | 9.65 | 3.53 | 56.35 | 38.52 | 19.65 |
| | MindEye+DINO | fMRI | 16.10 | 8.65 | 3.13 | 47.05 | 30.32 | 15.15 |
| | MindEye+CO-DINO | fMRI | 16.70 | 9.35 | 3.43 | 52.35 | 36.12 | 18.85 |
| All | MindEye+DAB-DETR | fMRI | 7.47 | 4.12 | 1.57 | 28.43 | 17.25 | 8.20 |
| | MindEye+DINO | fMRI | 7.07 | 3.92 | 1.47 | 24.13 | 14.45 | 6.50 |
| | MindEye+CO-DINO | fMRI | 7.27 | 4.02 | 1.57 | 26.43 | 16.25 | 7.60 |

Table 5: The results on different ROIs.

| Object Type | ROI | Average Precision ↑ | | | Average Recall ↑ | | |
|---|---|---|---|---|---|---|---|
| | | $AP_{30}$ | $AP_{50}$ | $AP_{70}$ | $AR_{30}$ | $AR_{50}$ | $AR_{70}$ |
| Small | V1 | 0.50 | 0.00 | 0.00 | 6.70 | 2.20 | 0.10 |
| | V2 | 1.10 | 0.10 | 0.00 | 7.70 | 2.30 | 0.20 |
| | V3 | 1.40 | 0.30 | 0.10 | 9.50 | 3.20 | 0.40 |
| | V4 | 2.10 | 0.50 | 0.20 | 13.80 | 4.60 | 0.60 |
| Medium | V1 | 1.70 | 0.50 | 0.20 | 26.10 | 12.60 | 3.10 |
| | V2 | 2.00 | 0.50 | 0.00 | 28.20 | 14.30 | 2.90 |
| | V3 | 4.10 | 1.70 | 0.10 | 33.00 | 17.80 | 3.40 |
| | V4 | 4.40 | 1.10 | 0.10 | 37.20 | 19.90 | 3.60 |
| Large | V1 | 4.10 | 2.70 | 1.30 | 50.30 | 40.20 | 21.30 |
| | V2 | 6.10 | 4.10 | 1.90 | 56.30 | 44.60 | 22.80 |
| | V3 | 8.70 | 5.90 | 2.60 | 63.60 | 51.60 | 27.50 |
| | V4 | 11.30 | 7.60 | 3.10 | 65.60 | 54.30 | 28.20 |
| All | V1 | 1.90 | 1.10 | 0.60 | 28.30 | 18.90 | 9.20 |
| | V2 | 2.80 | 1.80 | 0.80 | 31.10 | 21.20 | 10.00 |
| | V3 | 4.30 | 2.70 | 1.20 | 36.50 | 25.30 | 11.70 |
| | V4 | 5.70 | 3.50 | 1.40 | 40.30 | 27.50 | 12.30 |

To explore the role of visual regions (ROIs), we trained and tested using only the voxels from V1, V2, V3, and V4, respectively, yielding the results shown in the Tab. 5.

The number of voxels in V1, V2, V3, and V4 progressively decreases, with approximate ranges of 3000–4000, 2000–3000, 1500–2000, and 1000–1300, respectively. However, the detection results show an inverse trend, where higher visual areas (V3 and V4) outperform lower visual areas (V1 and V2). Additionally, the difference in the AR metric is smaller than that in the AP metric, as shown in the table. This indicates that the higher visual areas, V3 and V4, which undergo more advanced neural encoding, contain richer information and are easier to decode for object location and category. In contrast, while the lower visual areas, V1 and V2, also retain some information (as reflected by an acceptable AR metric), factors such as noise make decoding object location and category more challenging, resulting in a lower AP metric.

## A.8 COMPARISON OF MATCHING RATE UNDER DIFFERENT IOUS.

To measure the consistency in semantics and location, we performed the following calculations:

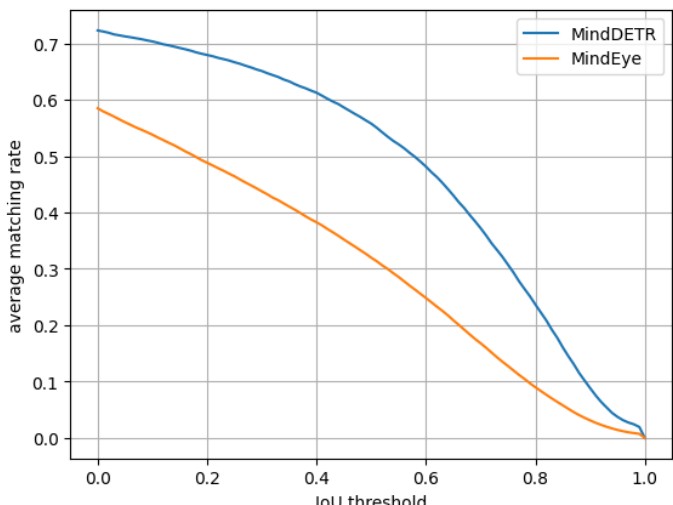

Figure 10: **The comparison of matching rate under different IoU.**

For a fixed IoU threshold, we analyzed the detection results of two different subjects on the same visual stimulus image. If a bounding box from one subject and a bounding box from another subject share the same category and their IoU meets or exceeds the threshold, the two bounding boxes are considered a match. We then averaged the matching rates across all subject pairs to obtain the final consistency score, as illustrated in Figure 10.

It can be seen that the detection stability of MindDETR is significantly higher than that of MindEye.

