# OpenReview forum: "MindDETR: Beyond Semantics, Exploring Positional Cues from Brain Activity"
_ICLR.cc/2025/Conference — Submitted to ICLR 2025_

### Official Review · Reviewer_MCRm · 2024-10-28

**Soundness:** 3
**Presentation:** 2
**Contribution:** 2
**Rating:** 6
**Confidence:** 4

**Summary:**

This paper presents a new task of object detection using fMRI data acquired while viewing images. The authors distill DETR, a pre-trained model in the object detection field so that the fMRI embedding can be more attentive to position information about the object in the scene. The object detection task header is then used to output the bounding boxes. The methods and evaluations seem sound.

**Strengths:**

+ Existing brain visual decoding focuses on classification, retrieval, and image reconstruction tasks; this paper focuses on the object detection task for the first time.
+ The experiments in the paper show that the proposed methods are effective and have yielded some results.

**Weaknesses:**

#### 1. The function of fMRI object detection requires further clarification
Although the authors have mentioned that fMRI target detection is potentially useful for vision-related neuroscience, however, it has only been talked about in general terms without really considering its use. As an innovative paper, it is particularly important to utilize a separate section to **discuss in detail the use of fMRI target detection for neuroscience or brain visual decoding**, which is relevant for judging the value of this paper.

#### 2. Lack analysis for limited performance
The accuracy of fMRI object detection is still low and the authors should further discuss the reasons for this result.

#### 3. Lack of analysis of error cases
Based on the quantitative results of the experiments, there should be many cases of incorrect detection, and the authors should analyze the incorrect cases rather than just showing the correct ones. A systematic analysis of the error cases would have been helpful for subsequent studies.

**Questions:**

+ Lines 400-406 state the proposed method "can maintain consistency in semantics, location, and quantity in most cases for the brain detection results of different subjects with the same visual stimulus images". I think this conclusion is meaningful, can the authors give quantitative results to further validate this conclusion?
+ Could the authors further provide more visualizations (corresponding to Figures 3, 4, 6) in the Appendix?

---

> ### Author Response · Authors · 2024-11-27
> **To Reviewer MCRm**
>
> W1:
> For brain visual decoding, pixel-level reconstruction/decoding is currently very challenging. We hope to provide some insights through the proxy task of determining object location/boundaries. In neuroscience, the significance of decoding tasks is quite limited, including previous reconstruction methods, which are more of an attempt and verification. There is also relatively little work on the encoding and decoding of location information. We hope to inspire other researchers.
>
> W2:
> Limited performance can be attributed to the NSD dataset’s lack of focus on object detection during its collection phase. This oversight means that many small and medium-sized objects in the visual stimuli may have been overlooked by participants, making it difficult to extract their location information from MRI data. Additionally, the dataset’s size is inadequate; it has been observed that a participant has seen no more than 10,000 images, which is typically insufficient for a model to develop robust generalization capabilities through training for more complex computer vision tasks. This limitation applies to our model as well, given its parameter count is comparable to that of DETR for visual object detection.
>
> W3:
> See 'To Reviewer 5Sff - W5'.
>
> Q1:
> To measure the consistency in semantics and location, we performed the following calculations:
>
> For a fixed IoU threshold, we analyzed the detection results of two different subjects on the same visual stimulus image. If a bounding box from one subject and a bounding box from another subject share the same category and their IoU meets or exceeds the threshold, the two bounding boxes are considered a match. We then averaged the matching rates across all subject pairs to obtain the final consistency score, as illustrated in Fig.10 in A.8.
>
> It can be seen that the detection stability of MindDETR is significantly higher than that of MindEye.
>
> Q2:
> See W3.

---

> > ### Comment · Reviewer_MCRm · 2024-11-27
> >
> > Thanks to your response, I have read your feedback carefully. However, I still have some questions:
> >
> > `W1a:` I think the significance of fMRI object detection in neuroscience is still vague. The authors have not clarified the paper's potential applications or insights for neuroscience. In fact, finding some studies that explore localization perception may enhance the persuasiveness of the argument.
> >
> > `W1b:` The significance of fMRI object detection in fMRI visual decoding is still unclear. The authors have recognized that fMRI object detection is only a proxy task, so this paper should at least make some preliminary inquiry into how the bounding boxes of localized objects can be used to alleviate the problem of inaccurate object position during visual reconstruction. This is **particularly important** as this was the motivation for this study and was noted by all reviewers.
> >
> > For these reasons, I think this paper may not have contributed as much as I expected. So I decided to lower “Contribution” from 3 to 2.

---

### Official Review · Reviewer_T78L · 2024-10-31

**Soundness:** 3
**Presentation:** 3
**Contribution:** 2
**Rating:** 5
**Confidence:** 4

**Summary:**

This paper finds that existing methods struggle to accurately recover object positional information. To address this issue, the authors incorporate an object detection task into brain signal decoding and propose a new method named MindDETR. Experimental results demonstrate the feasibility of performing object detection during brain decoding.

**Strengths:**

This paper conduct a new task, where conduct object detection based on fMRI signals.

This paper proposed an interesting method, MindDETR, and the performance on object detection beat all the baselines on all kinds of object types.

**Weaknesses:**

1. First of all, I believe the proxy task of object detection should not be a standalone task; rather, it should serve as an auxiliary task to enhance brain decoding performance. The images in the participant's mind may differ significantly from those they view.

2. Comparing the bounding boxes predicted by MindDETR on ground truth images with those predicted by baseline models on reconstructed images is quite unfair. I suggest that the authors compare the performance of all models (except for the standard object detection model) on reconstructed images using the predicted bounding boxes.

3. In line 310, please explain in detail how object detection is conducted on these reconstructed images. Which object detection model did you use? This clarification may help address any concerns about the fairness of the experimental results.

4. Brain decoding primarily relies on visual regions (ROIs) in the human brain, but I believe object location may depend on certain cognitive regions. I suggest some exploration in this area.

**Questions:**

Refer to the weaknesses.

---

> ### Author Response · Authors · 2024-11-27
> **To Reviewer T78L-Part1**
>
> We thank Reviewer T78L for his/her valuable feedback, and here we provide corresponding responses to address these concerns.
>
> W1:
> We totally agree with these two comments.
> Given that the mental images of participants may significantly differ from what they perceive visually, achieving an exact reconstruction of the original image is inherently challenging.
> Consequently, we have introduced this proxy task, aiming to decode MRI signals from alternative perspectives.
> We also believe that the detection task can work with reconstruction task (as discussed in Section 5.2 limitation and future work). For example, by incorporating positional information as an additional condition, reconstruction accuracy could be further improved, as suggested in related works [1, 2].
>
> References:
>
> [1] Feng, Yutong, et al. "Ranni: Taming text-to-image diffusion for accurate instruction following." Proceedings of the IEEE/CVF Conference on Computer Vision and Pattern Recognition. 2024.
>
> [2] Yun, Jooyeol, et al. "Generative Location Modeling for Spatially Aware Object Insertion." arXiv preprint arXiv:2410.13564. 2024.
>
> W2, W3:
>
> Here is the detailed explanation of line 310:
>
> Firstly, we resize the reconstructed images from Takagi and MindEye to a standard size of 425x425 pixels, which matches the visual stimulus' size in the NSD dataset. Then we conduct object detection using DAB-DETR on these reconstructed images. Before detection, to align the dimensions of the input image, the images will be reshaped to 800 * 800 pixels. After detection, the detected bounding boxes will be used to calculate the metrics with the annotated bounding boxes of the original data stimulus images. The calculation rules for the metric are consistent with COCO.
>
> We have incorporated two additional models, DINO[1] and CO-DINO[2], for detecting the reconstructed images, which exhibit better detection performance on small objects. The results are shown in Tab.4 in A.6.
>
> As shown in the table, the choice of detection model has a noticeable impact on the results. However, a stronger detection model does not always yield better performance and may even cause slight metric regression. We hypothesize that this is because stronger detection models are more likely to identify blurred objects in the reconstructed images generated by MindEye, which weaker models might overlook. The inaccurate positional information associated with these blurred objects could contribute to the observed decline in metrics.
>
> References:
>
> [1] Zhang, Hao, et al. "DINO: DETR with Improved DeNoising Anchor Boxes for End-to-End Object Detection." The Eleventh International Conference on Learning Representations.
>
> [2] Zong, Zhuofan, Guanglu Song, and Yu Liu. "Detrs with collaborative hybrid assignments training." Proceedings of the IEEE/CVF international conference on computer vision. 2023.

---

> > ### Comment · Reviewer_T78L · 2024-11-29
> >
> > Dear authers, thank you very much for your valuable effort and the well-prepared response, which has addressed most of my concerns. However, there is still an important concern: the authors should compare the detection results on the reconstructed images from the proposed method rather than on the ground truth images. I am not concerned with the performance of the detection model itself; as long as the same detector is used across all models, the comparison is fair. However, comparing detection on ground truth versus reconstructed images is inherently unfair. Therefore, I will keep my score as 5.

---

> ### Author Response · Authors · 2024-11-27
> **To Reviewer T78L-Part2**
>
> W4:
> Thanks for this suggestion.
> To explore the role of visual regions (ROIs), we trained and tested using only the voxels from V1, V2, V3, and V4, respectively, yielding the results shown in the following table:
>
> | Object Type | ROI  | AP30 | AP50 | AP70 | AR30 | AR50 | AR70 |
> |-------------|------|-------|-------|-------|-------|-------|-------|
> | Small       | V1   | 0.50  | 0.00  | 0.00  | 6.70  | 2.20  | 0.10  |
> | Small       | V2   | 1.10  | 0.10  | 0.00  | 7.70  | 2.30  | 0.20  |
> | Small       | V3   | 1.40  | 0.30  | 0.10  | 9.50  | 3.20  | 0.40  |
> | Small       | V4   | 2.10  | 0.50  | 0.20  | 13.80 | 4.60  | 0.60  |
> | Medium      | V1   | 1.70  | 0.50  | 0.20  | 26.10 | 12.60 | 3.10  |
> | Medium      | V2   | 2.00  | 0.50  | 0.00  | 28.20 | 14.30 | 2.90  |
> | Medium      | V3   | 4.10  | 1.70  | 0.10  | 33.00 | 17.80 | 3.40  |
> | Medium      | V4   | 4.40  | 1.10  | 0.10  | 37.20 | 19.90 | 3.60  |
> | Large       | V1   | 4.10  | 2.70  | 1.30  | 50.30 | 40.20 | 21.30 |
> | Large       | V2   | 6.10  | 4.10  | 1.90  | 56.30 | 44.60 | 22.80 |
> | Large       | V3   | 8.70  | 5.90  | 2.60  | 63.60 | 51.60 | 27.50 |
> | Large       | V4   | 11.30 | 7.60  | 3.10  | 65.60 | 54.30 | 28.20 |
> | All         | V1   | 1.90  | 1.10  | 0.60  | 28.30 | 18.90 | 9.20  |
> | All         | V2   | 2.80  | 1.80  | 0.80  | 31.10 | 21.20 | 10.00 |
> | All         | V3   | 4.30  | 2.70  | 1.20  | 36.50 | 25.30 | 11.70 |
> | All         | V4   | 5.70  | 3.50  | 1.40  | 40.30 | 27.50 | 12.30 |
>
> The number of voxels in V1, V2, V3, and V4 progressively decreases, with approximate ranges of 3000–4000, 2000–3000, 1500–2000, and 1000–1300, respectively. However, the detection results show an inverse trend, where higher visual areas (V3 and V4) outperform lower visual areas (V1 and V2). Additionally, the difference in the AR metric is smaller than that in the AP metric, as shown in the table. This indicates that the higher visual areas, V3 and V4, which undergo more advanced neural encoding, contain richer information and are easier to decode for object location and category. In contrast, while the lower visual areas, V1 and V2, also retain some information (as reflected by an acceptable AR metric), factors such as noise make decoding object location and category more challenging, resulting in a lower AP metric.

---

### Official Review · Reviewer_5Sff · 2024-11-04

**Soundness:** 2
**Presentation:** 3
**Contribution:** 2
**Rating:** 3
**Confidence:** 2

**Summary:**

This paper focuses on the task of brain signal decoding. Instead of adopting generative models to reconstruct images from brain signals, this paper proposes to use object detection as the proxy task, to decode both semantic and positional cues from brain recording. To this end, it presents MindDETR, a brain recording-based object detection model to align feature representations with a pre-trained image-based DETR model. Experiments are conducted to evaluate the effectiveness of the proposed method.

**Strengths:**

1. The task of this paper is interesting. The paper is well-written and easy to understand.
2. Extensive experiments are conducted on NSD dataset, showing the superiority of the proposed method to other competitors.

**Weaknesses:**

1.	The motivation of this paper is ambiguous to me.
 + Is it proposed to tackle the issue of inaccurate positional features among brain signal decoding (as stated in Sec. 1 and Sec. A.2)? It is a very good question, but instead of solving this issue, this paper raises a new brain decoding task about object detection.
 + Can the proposed proxy task of object detection promote positional awareness during reconstructing visual stimulus from fMRI signals? This paper doesn’t discuss this point.
 + Is it proposed to reveal that the fMRI signals from NSD dataset can do the detection task (claimed as the second contribution in L95)? From the experimental results, the detection performance is not satisfactory (e.g., 8.5 AP_{50} for all objects)

2. The proposed framework primarily uses non-linear functions to map brain signal embedding to image features, to fulfill the task of object detection. From the perspective of computer vision, this is a common operation for feature alignment, especially in the current era of multimodal tasks. IMO, the quality of feature alignment depends on the validity or consistency of two features. Thus, my concern is whether the fMRI signals from NSD datasets are really suitable for object detection or whether brain activity should be collected while asking subjects to do something like object detection. This point is not well discussed or analyzed in depth by the authors, and the experimental results (i.e., metrics for object detection) are not good. I understand that it may be beyond the scope of this paper (more like the field of brain neuroscience), but without this assumption or premise, the significance of this paper is very unclear.

3. The technical novelty of this proposed method is limited. Both the usage of low- and high-level features and the feature alignment via distillation have been proposed by previous literature [A,B,C]. In addition, why does the larger kernel size yield better results? The author didn’t provide any explanation in Sec. 3.3 or 4.4. It is more of an experimental hyper-parameter selection, with no basis.

4. In Tab. 1, the proposed MindDETR achieves better results than MindEye. Could the author briefly describe the implementation of MindEye for object detection, and also state the architectural differences with the proposed method?

5. Are the visualizations in Fig.3 cherry-picked? As seen from Tab. 2, the precision and recall of the proposed method actually are not high. It would be better to provide in-depth discussions of the failure cases.

6. The discussion about Fig. 4 is not convincing. First of all, MindEye is for image reconstruction and MindDETR is for object detection. Two results are not comparable. Second, the proposed MindDETR also shows differences among different subjects, including the position, scale, and confidence of bounding boxes. Especially for multiple objects, the difference between individual results will be more obvious (like Fig.4 (e) and (f)). Besides, the subtitle of ‘Consisteny among different objects’ should be ‘Consistency among different subjects’.

[A] Scotti, Paul, et al. "Reconstructing the mind's eye: fMRI-to-image with contrastive learning and diffusion priors." Advances in Neural Information Processing Systems 36 (2024)

[B] Ozcelik, Furkan, and Rufin VanRullen. "Natural scene reconstruction from fMRI signals using generative latent diffusion." Scientific Reports 13.1 (2023): 15666

[C] Liu, Yulong, et al. "BrainCLIP: Bridging brain and visual-linguistic representation via CLIP for generic natural visual stimulus decoding." arXiv preprint arXiv:2302.12971 (2023).

**Questions:**

Please see the Weaknesses for details.

---

> ### Author Response · Authors · 2024-11-27
> **To Reviewer 5Sff - Part1**
>
> Thanks for your valuable feedback, and here we provide corresponding responses to address these concerns.
>
> W1:
> Our motivation stems from addressing the challenge of inaccurate positional features in brain signal decoding during image reconstruction tasks. However, this is an exceedingly complex problem, and we aim to inspire and encourage the research community to engage with this promising research direction by introducing this proxy task.
>
> First, as shown in the experimental results, the proposed proxy task optimizes the models based on the positional-aware objective, thus improving the accuracy of position decoding.
> Second, the brain detection results obtained through our proxy task have the potential to enhance positional awareness in reconstruction task. Current reconstruction methods generally use decoded CLIP embeddings as conditions for Stable Diffusion models. By incorporating positional information as an additional condition, reconstruction accuracy could be further improved, as suggested in related works [1, 2].
>
>
> First, although the AP50 metric for this task is relatively low, this is primarily due to the presence of a large number of small objects in the NSD dataset (sourced from COCO), which reduces the average localization accuracy. These small objects are often overlooked in the primary evaluation metrics (semantic consistency) for image reconstruction tasks.
> Second, our preliminary exploration demonstrates that detection-based methods significantly outperform reconstruction-based methods in terms of localization accuracy, indicating that the quality of the existing dataset can support brain decoding with higher localization precision.
> Third, the current approach is a baseline model for future work, and we believe there is substantial room for improvement.
>
> References:
>
> [1] Feng, Yutong, et al. "Ranni: Taming text-to-image diffusion for accurate instruction following." Proceedings of the IEEE/CVF Conference on Computer Vision and Pattern Recognition. 2024.
>
> [2] Yun, Jooyeol, et al. "Generative Location Modeling for Spatially Aware Object Insertion." arXiv preprint arXiv:2410.13564. 2024.
>
> W2:
> During the collection of the NSD dataset, subjects were asked to observe and recall the details of the images as much as possible. At the same time, subjects were fixed during observation, so the positional information of the objects exists.
> Although the current results remain relatively low compared to image-based detectors, the proposed task and baseline model demonstrate improved fidelity in spatial information decoding over existing reconstruction methods. This indicates that the potential of spatial information decoding in the current NSD dataset has yet to be fully exploited. At the same time, we acknowledge that more systematic data collection strategies—such as requiring participants to pre-memorize location-specific features—could further advance progress in this field.
>
> W3:
> The role of the convolution kernel in our model is similar to that in the visual CNN backbone. Generally, smaller convolution kernels (such as 3x3 or 5x5) are often used to capture local features, while larger convolution kernels (such as 7x7) may be used to capture broader contextual information. In MindDETR, we found that the generalization ability of the linear layer is insufficient, so we added a convolutional layer to extract local features. Determining the size of the convolution kernel through experimentation is a common practice in the design of visual models.
>
> We also acknowledge that both object detection and cross-modal distillation are established techniques in deep learning. However, the contribution we would like to emphasize more is: prior image-reconstruction based methods have largely overlooked the evaluation of spatial information in their metrics, with a focus of the semantic consistency of the main object. This omission may lead researchers in the field to misjudge the fidelity of brain signal decoding. In this work, we build a proxy task and provide a baseline model to encourage the future works on this field.
>
> W4:
> MindEye focuses on image reconstruction and does not directly provide object location information. To address this, we used a pre-trained object detection model to detect objects in the images reconstructed by MindEye, enabling us to extract object location data from the reconstructions.
>
> Notably, during training, MindEye did not utilize bounding box information from the dataset. Its primary objective is to predict or reconstruct the CLIP visual embeddings of visual stimuli. The core of MindEye's training process relies on a contrastive loss, which emphasizes semantic consistency of embeddings. In contrast, our model is specifically designed to decode location information, enabling it to achieve better performance in this regard compared to MindEye.

---

> > ### Author Response · Authors · 2024-11-27
> > **To Reviewer 5Sff - Part2**
> >
> > W5:
> > In the main paper, we mainly visualized several large objects to more clearly demonstrate the differences between our approach and image reconstruction-based methods. We are also willing to provide some failure cases with corresponding analyses.
> >
> > We present more failure cases in Figure 9 (Appendix A5 in the revised pdf file):
> >
> > Case (a): Small objects like spoons, knives, donuts, and ovens were not detected, and persons appearing at the image edges were sometimes missed by some subjects. This raises the question of whether detection on the NSD dataset should include such small objects. Decoding fine details that subjects are less likely to notice from MRI signals is inherently challenging. By contrast, decoding central objects appears to be easier, while edge objects are more prone to being overlooked.
> >
> > Case (b): A clock at the center of the image was not detected due to weak visual stimulation. The primary focus of the visual stimulus was architecture, not the clock. Since the clock category often functions as a non-core semantic object, it may be worth reconsidering whether this category should be included in the detection process.
> >
> > Case (c): A prominent object, such as a kite, was sometimes not detected, which is unexpected as it should be noticed by all subjects. This variability suggests that the model's learning for this category remains inadequate, potentially due to differences in how subjects perceive or prioritize the kite.
> >
> > Case (d): The detection of a person in the background was highly blurred, indicating that background objects are harder to detect, even though they may not pose a challenge for image-based detection. Similarly, the core semantic object, a baseball bat, was not detected, likely due to its small size.
> >
> > In summary, objects located in the background, at the edges of images, or with small sizes tend to be more difficult to detect.
> >
> > W6:
> > In the section “Consistency among Different Objects,” we aim to highlight that the instability of the diffusion process results in significant randomness in the reconstructed object positions, which partially explains the suboptimal detection performance. Figure 7 demonstrates this by using the MRI scan results of the same subject as input, paired with different random seeds, to perform multiple independent reconstructions. Our model, however, provides consistent object position results.
> >
> > As further illustrated in Figures 4(e) and 4(f), decoding results for the same visual stimulus may vary across subjects, which is expected due to individual differences in neural responses to the same stimulus. Nonetheless, in terms of decoding positional information, our method clearly outperforms the reconstruction approach.

---

### Author Response · Authors · 2024-11-27
**The appendix section of the new PDF file has been updated with additional content pertaining to the rebuttal.**

We sincerely thank all the reviewers for their hard work and valuable feedback. The appendix section of the new PDF file has been updated with additional content pertaining to the rebuttal. We kindly ask all reviewers to take a look.

---

### Meta-Review · Area_Chair_T3dt · 2024-12-21

**Metareview:**

This work introduces MindDETR, a framework leveraging fMRI data for object detection as a novel proxy task to improve brain signal decoding with positional information. While the idea of using object detection for decoding positional cues is intriguing, reviewers identified significant concerns. Most notably, there is a lack of clarity and fairness in the comparison of bounding boxes between ground truth images and reconstructed images from baseline models, seriously questioning the validity of performance claims. Additionally, reviewers pointed out limited discussion on the significance of the bounding boxes for visual reconstruction tasks and the absence of systematic error analysis for failure cases. These issues, coupled with questions about the suitability of the fMRI dataset for this task and the technical novelty of the proposed approach, suggest further improvement.

**Additional Comments On Reviewer Discussion:**

Two reviewers have raised concerns regarding the validity and fairness of comparing bounding box predictions on ground truth images versus reconstructed images, the limited technical novelty of the approach, and the unclear significance of the proxy task of object detection for neuroscience and brain decoding applications. The authors responded by clarifying the detection pipeline and experimental setup, providing additional visualizations and failure case analyses, and expanding on the potential utility of object detection for improving positional awareness in reconstruction tasks. While these efforts addressed some concerns, the fundamental issues—such as the unclear role of bounding boxes in improving visual decoding and the unfair experimental comparisons—remained unresolved. These unresolved concerns, along with limited methodological innovation, suggesting a rejection.

---

### Decision · Program_Chairs · 2025-01-22

Reject